# Vapor-Induced Pore-Forming Atmospheric-Plasma-Sprayed Zinc-, Strontium-, and Magnesium-Doped Hydroxyapatite Coatings on Titanium Implants Enhance New Bone Formation—An In Vivo and In Vitro Investigation

**DOI:** 10.3390/ijms24054933

**Published:** 2023-03-03

**Authors:** Hsin-Han Hou, Bor-Shiunn Lee, Yu-Cheng Liu, Yi-Ping Wang, Wei-Ting Kuo, I-Hui Chen, Ai-Chia He, Chern-Hsiung Lai, Kuo-Lun Tung, Yi-Wen Chen

**Affiliations:** 1Graduate Institute of Oral Biology, School of Dentistry, National Taiwan University, Taipei 10048, Taiwan; 2Department of Dentistry, National Taiwan University Hospital, Taipei 10048, Taiwan; 3Graduate Institute of Clinical Dentistry, School of Dentistry, National Taiwan University, Taipei 10048, Taiwan; 4Department of Chemical Engineering, National Taiwan University, Taipei 10617, Taiwan; 5Division of Periodontology, Department of Dentistry, Kaohsiung Medical University Hospital, Kaohsiung 80756, Taiwan; 6College of Life Science, Kaohsiung Medical University, Kaohsiung 80756, Taiwan

**Keywords:** atmospheric plasma, hydroxyapatite coating, osteogenesis, antibacterial, dental implant, zinc, strontium, magnesium

## Abstract

Objectives: Titanium implants are regarded as a promising treatment modality for replacing missing teeth. Osteointegration and antibacterial properties are both desirable characteristics for titanium dental implants. The aim of this study was to create zinc (Zn)-, strontium (Sr)-, and magnesium (Mg)-multidoped hydroxyapatite (HAp) porous coatings, including HAp, Zn-doped HAp, and Zn-Sr-Mg-doped HAp, on titanium discs and implants using the vapor-induced pore-forming atmospheric plasma spraying (VIPF-APS) technique. Methods: The mRNA and protein levels of osteogenesis-associated genes such as collagen type I alpha 1 chain (COL1A1), decorin (DCN), osteoprotegerin (TNFRSF11B), and osteopontin (SPP1) were examined in human embryonic palatal mesenchymal cells. The antibacterial effects against periodontal bacteria, including *Porphyromonas gingivalis* and *Prevotella nigrescens*, were investigated. In addition, a rat animal model was used to evaluate new bone formation via histologic examination and micro-computed tomography (CT). Results: The ZnSrMg-HAp group was the most effective at inducing mRNA and protein expression of TNFRSF11B and SPP1 after 7 days of incubation, and TNFRSF11B and DCN after 11 days of incubation. In addition, both the ZnSrMg-HAp and Zn-HAp groups were effective against *P. gingivalis* and *P. nigrescens*. Furthermore, according to both in vitro studies and histologic findings, the ZnSrMg-HAp group exhibited the most prominent osteogenesis and concentrated bone growth along implant threads. Significance: A porous ZnSrMg-HAp coating using VIPF-APS could serve as a novel technique for coating titanium implant surfaces and preventing further bacterial infection.

## 1. Introduction

Titanium dental implants have revolutionized dentistry and become one of the most common treatment options for replacing missing teeth in partially or fully edentulous patients [1]. Osseointegration is defined as the direct functional and structural connection between living bone and the load-bearing surface of a titanium implant [2]. Osseointegration includes primary stability (mechanical stability) and secondary stability (biological stability). Mechanical stability is determined by the bone density and implant design, whereas biological stability is associated with physiologic bone healing. The cellular and molecular phenomena that occur during osseointegration form a cascade including blot clot formation, angiogenesis, the migration of osteoprogenitor cells, woven bone formation, and bone remodeling [3]. Therefore, the surface treatment of a dental implant may influence bone deposition and determine the implant success at an early stage. Various studies have proposed increasing surface hydrophilicity and attracting osteoprogenitor cells using the advantages of metal ions [4,5,6,7,8]. Treating titanium implant surfaces with calcium phosphate deposition via immersion in simulated body fluid under physiological conditions of temperature (37 °C) and pH (7.4) has been proven to enhance surface hydrophilicity, attract osteoinductive agents, and promote bone healing [9]. Strontium (Sr) has been reported to regulate osteoblast-related gene expression, enhance alkaline phosphatase (ALP) activity, and reduce osteoclast differentiation [5]. Strontium ranelate has been demonstrated to increase runt-related transcription factor 2 (Runx2) expression and matrix mineralization and attenuate bone resorption in an osteopenic mouse model [6]. Zinc (Zn) and magnesium (Mg) are essential elements for increasing alkaline phosphatase activity [7,8] and bone protein synthesis [10], thus promoting bone formation [11].

Peri-implantitis is defined as soft tissue inflammation and progressive bone loss around dental implants and is considered a polymicrobial anaerobic infection associated with biofilms [12,13]. Various clinical regimes for preventing and treating peri-implantitis have been proposed based on its pathophysiology, including mechanical debridement, laser treatment, locally delivered antiseptics, local or systemic antibiotics, and surgical access and regenerative procedures. However, a gold standard protocol for treating peri-implantitis has yet to be established [14]. Metal coatings, such as Ag^+^, Zn^2+^, Sr^2+^, Mg^2+^, Ca^2+^, F^−1^ and Sc^+3^, and antimicrobial peptides have been reported to inhibit bacterial adhesion and enhance osseointegration [15,16,17,18,19]. Hydroxyapatite (HAp) coatings doped with 1 wt% AgNO_3_ (AgHA1.0) exhibit the ability to minimize the initial adhesion of *Streptococcus aureus* and *Staphylcoccus epidermidis* [20]. Metal ions can prevent the emergence of drug-resistant bacteria resulting from the overuse of antibiotics. Titanium plates coated with a copper-HAp composite via two-stage electrochemical synthesis have demonstrated excellent antibacterial properties against *Escherichia coli* (Gram-negative) and *S. aureus* (Gram-positive) [21]. Moreover, Zn has been added to toothpaste and mouth rinses to inhibit calculus deposition and the growth of cariogenic bacteria [22]. Therefore, the development of an implant surface with antimicrobial properties is essential.

Implant coating materials, such as metals (titanium and its alloys, aluminum alloys, cobalt, and zirconium), ceramics (HAp), and polymers (polyurethane and polyethylene), are crucial to maintaining superior mechanical properties, corrosion resistance, and antimicrobial properties [23]. Surface modification treatments, including physical (electron beam evaporation, thermal spraying, pulsed laser deposition, and thermal evaporation) and chemical methods (chemical vapor deposition, electrophoretic deposition, and sol–gel coating), have been widely applied to improve surface properties [23]. HAp is among the most in-demand materials for the modification of surface properties for optimal osseointegration in implantology [24,25,26]. Furthermore, HAp is resistant to X-ray and UV irradiation and does not display visible aging/structural damage [27,28]. Our previous study proved that the vapor-induced pore-forming atmospheric plasma spraying (VIPF-APS) technique could effectively produce a porous HAp coating and contribute to a more bioactive coating for osteoblast proliferation [29]. Sr- and Mg-doped HAp implants were demonstrated to enhance osteoblast proliferation and new bone formation in a beagle dog model [29]. To the best of our knowledge, the effect of Zn-, Sr-, and Mg-multidoped HAp-coated titanium implants on the enhancement of bone formation has yet to be investigated. In addition, Zn-, Sr-, and Mg-multidoped HAp coatings produced using the VIPF-APS technique have never been examined. Therefore, the novelty of this study lies in the use of the VIPF-APS technique to create Zn-, Sr-, and Mg-multidoped HAp porous coatings on titanium discs and implants. The mRNA and protein levels of osteogenesis-associated genes such as collagen type I alpha 1 chain (COL1A1), decorin (DCN), osteoprotegerin (TNFRSF11B), and osteospontin (SPP1) were examined. The antibacterial effects against periodontal bacteria, including *Porphyromonas gingivalis* and *Prevotella nigrescens*, were investigated. Moreover, a rat model was used to evaluate osseointegration via histologic examination and micro-computed tomography (micro-CT).

## 2. Results

### 2.1. ZnSrMg-HAp Promotes Osteointegration-Associated Gene and Protein Expression in HEPM Cells

The HEPM cells were incubated on titanium discs coated with HAp, Zn-HAp, or ZnSrMg-HAp for 3 (Figure 1A), 7 (Figure 1B), and 11 days (Figure 1C). The purified cellular total RNA was used for a qPCR assay with primer sets of COL1A1, DCN, TNFRSF11B, and SPP1. The mRNA levels of TNFRSF11B and SPP1 significantly increased in the ZnSrMg-HAp group after 7 days of incubation compared with those of the HAp group (Figure 1B). Moreover, the mRNA levels of TNFRSF11B and DCN also significantly increased in the ZnSrMg-HAp group after 11 days of incubation compared with those of the HAp group (Figure 1C). To confirm the protein expression level according to the previous qPCR results, the cellular lysate was used in a Western blot assay. The protein levels of TNFRSF11B and SPP1 significantly increased in the ZnSrMg-HAp group after 7 days of incubation compared with those of the HAp group (Figure 2A,B). Moreover, the protein levels of TNFRSF11B and DCN significantly increased in the ZnSrMg-HAp group after 11 days of incubation compared with those of the HAp group (Figure 2C,D). These results demonstrated that the titanium discs coated with ZnSrMg-HAp prominently increased the expression of osteointegration-associated genes and proteins and suggested that the ZnSrMg-HAp coating promoted osteointegration ability.

### 2.2. Antibacterial Activity Test of HAp, Zn-HAp, and ZnSrMg-HAp Coatings on Titanium Discs against P. gingivalis and P. nigrescens

The ZnSrMg-HAp group demonstrated superior antibacterial activity against *P. gingivalis* compared with the Zn-HAp and HAp groups (Figure 3A). The ZnSrMg-HAp and Zn-HAp groups demonstrated prominent antibacterial activity against *P. nigrescens*. However, no difference was observed between the ZnSrMg-HAp and Zn-HAp groups (Figure 3B). In contrast, the HAp group did not demonstrate an apparent antibacterial effect against *P. gingivalis* or *P. nigrescens*.

### 2.3. Micro-CT Assessment

Micro-CT reconstruction images are shown in Figure 4A. Within the ROI of 0.85 mm and 1.1 mm, the bone coverage rate was significantly higher in the Zn-HAp and ZnSrMg-HAp groups compared to the HAp group at 2 and 4 weeks (Figure 4B,C). BV/TV was also significantly higher in the Zn-HAp and ZnSrMg-HAp groups at 2 and 4 weeks compared to the HAp group (Figure 4D). In contrast, the Zn-HAp and ZnSrMg-HAp groups exhibited significantly lower BMDs at 2 and 4 weeks compared to the HAp group (Figure 4E).

### 2.4. Histological Findings

Figure 5 shows that the ZnSrMg-HAp group exhibited better osteointegration than the HAp and Zn-HAp groups. At 2 weeks after implantation, the HAp group only achieved integration within the cortical bone area (the first and second threads). A focal discontinuous deposit of the bone on the implant surface was observed down to the interthread area between the third and fourth threads. Similar effects were also discerned in the Zn-HAp group, with a patchy surface bony deposit present down to the region slightly beyond the tip of the fifth thread (the third thread in the cancellous bone). In contrast, continuous bone formation on the implant surface was evident down to the lower slope of the sixth thread in the ZnSrMg-HAp group. The advantage of ZnSrMg-HAp surface processing in terms of osteointegration was more profound 4 weeks after implantation. Continuous and complete bone coverage was identified on all threads in the cancellous bone in the ZnSrMg-HAp group, whereas this feature could only be found down to the fourth and fifth threads in the HAp and Zn-HAp groups, respectively. At higher magnification (100×, thickened bone deposition was more apparent on the implant surface in the ZnSrMg-HAp group than in the HAp and Zn groups.

## 3. Discussion

This was the first study to use VIPF-APS to prepare porous coatings on dental implant surfaces. In addition, three-element doped HAp was successfully produced using ion doping technology. Compared with multi-element doped HAp prepared using the traditional solid-phase method [30], the coprecipitation method adopted in this study controlled the amount of ion doping more precisely and suppressed the generation of other non-targeted substances resulting from HAp phase transformation or an incomplete oxidative process [31]. Various coating techniques, such as plasma spraying, hydrocoating, two-stage processing, physical vapor deposition, thermally applied coating, and nanoscale technology, have been developed for implant surface modification [32]. Plasma spraying is the most commonly used technique for applying ion coatings on implant surfaces. The physical principles of the novel VIPF-APS technique in this study are based on the penetration of expansive vapors through melted HAp. The porous structure serves as a cavity for the recruitment of undifferentiated mesenchymal cells to the implant surface compared with the dense HAp coatings using traditional APS. Achieving a balance between mechanical strength and adequate pore size for bone ingrowth is critical for implant surface modification. In this study, the pore diameter of the coatings on titanium discs prepared via VIPF-APS was approximately 38 μm [33]. Our previous study demonstrated that the VIPF-APS technique could effectively produce porous HAp coatings that are favorable for higher osteoblast proliferation and alkaline phosphatase activity. The size of interconnecting pores is critical in affecting bone ingrowth. A pore size between 100 and 400 μm has been suggested to be favorable for bone ingrowth [34]. Another study showed that a pore size range of approximately 50 to 400 μm is optimal for fixation strength (17 MPa) [35]. The HAp coatings prepared using the VIPF-APS technique had more than 8% porosity compared to the traditional APS technique [31]. Therefore, we used the advantage of the VIPF-APS technique to produce Zn-, Sr-, and Mg-doped HAp coatings on titanium discs and implants. In addition to the successful preparation of porous HAp coatings on dental implants, this study also succeeded in preparing a three-element doped HAp coating powder for VIPF-APS.

HEPM cells are preosteoblasts that can differentiate into osteoblasts on titanium plates [36]. The growth and differentiation of osteoblasts can be divided into three periods based on the different genes expressed in each period: proliferation, extracellular matrix maturation, and extracellular matrix mineralization. First, collagen 1 and 2 (COL1, 2) are upregulated in the early stages of osteoblast differentiation. Next, SPP1 is required for extracellular matrix maturation. Finally, in the period of extracellular matrix mineralization, Ca^2+^ binding proteoglycans such as biglycan and DCN are secreted [37]. In addition, DCN has been proven to modulate collagen matrix assembly and mineralization [38] and regulate the cell cycle [39]. Otherwise, TNFRSF11B, a tumor necrosis factor receptor superfamily member, functions as a negative regulator of bone resorption by regulating osteoclast development [40]. Thus, we quantified the mRNA and protein expression of gene markers, including COL1, SPP1, DCN, and TNFRSF11B at different stages of HEPM (3, 7 and 11 days). The results showed that Zn-HAp had significantly higher COL1 and DCN mRNA than the HAp group on the third day. On the 11th day, Zn-HAp and ZnSrMg-HAp had significantly higher DCN and TNFRSF11B activity than HAp. In a previous study, human dental pulp stem cells cultured on Zn-modified titanium plates were proven to enhance the expression of osteoblast-related genes, such as COL1, bone morphogenetic protein 2, ALP, Runx2, osteopontin, and vascular endothelial growth factor A in vitro [41]. Our results were in agreement with previous studies that demonstrated the upregulating ability of Zn in an osteoblast culture (24–72 h) [42,43]. In addition, ZnSrMg-HAp had significantly higher SPP1 and TNFRSF11B activity than the HAp group on the seventh day. In bone-defect mice, the promoter activity of nuclear factor-kappa beta and vascular endothelial growth factor receptor-2 is upregulated by Sr supplementation [44]. Mg supplementation has been proven to upregulate the mRNA of peroxisome proliferator-activated receptor gamma and glucose transporter 1 in peripheral blood mononuclear cells from women with gestational diabetes [4]. The results of this study were in line with the process of bone differentiation and proved that Zn alone or ZnSrMg were effective in improving bone growth and maturation. In other words, ZnSrMg plays a critical role in the process of osteoblast growth and differentiation in vitro.

Antibiotics represent the most effective method for treating peri-implantitis in a clinical setting. However, bacterial resistance could be a concern because of excessive antibiotic usage [45]. To examine the effect of element-doped HAp coatings against periodontal pathogens, we chose *P. gingivalis* and *P. nigrsecens* as the target bacteria. Our results showed that the antibacterial activity of the ZnSrMg-HAp group was superior to that of the Zn-HAp group. Previous studies have reported that Zn, Sr, and Mg ions released from HAp coatings enhanced bone mineralization and exhibited antibacterial properties [46,47,48]. The release of mental ions from HAp coatings formed an alkaline environment and was not favorable for bacterial growth. Moreover, the membrane potential difference caused by metal ions may result in electron transfer and generate excessive amounts of reactive oxygen species, which further kills bacteria [49]. An implant surface with antibacterial properties, particularly the inhibition of biofilm formation and bacterial adhesion, is the most promising strategy for preventing or treating peri-implantitis.

Unlike the results regarding protein expression shown in Figure 2, the bone coverage rate in the Zn-HAp group was not significantly different from that of the ZnSrMg-HAp group. The reason might be that animals were sacrificed after at least 2 weeks, which was longer than the examination time for protein expression. The bone growth for osteointegration occurs very early after implant placement. The ZnSrMg-HAp group exhibited a higher bone coverage rate at an ROI of 0.85 mm compared with the Zn-HAp group at both 2 and 4 weeks (Figure 4B). In contrast, the bone coverage rate was higher in the Zn-HAp group than in the ZnSrMg-HAp group at an ROI of 1.1 mm (Figure 4C). However, no significant difference was found for both ROIs. The Sr^2+^ and Mg^2+^ ions released from ZnSrMg-HAp could stimulate osteogenesis and new bone formation concentrated along the implant threads. Therefore, the bone coverage rate of the ZnSrMg-HAp group was more prominent at the smaller ROI (0.85 mm). These results were in agreement with the histological findings, which demonstrated that the ZnSrMg-HAp group exhibited more prominent continuous bone coverage on implant threads compared with the HAp and Zn-HAp groups (Figure 5). In addition, the concentrated bone growth on implant threads observed during the histological examination was not composed of cancellous bone. The bone volume fraction (BV/TV) did not differ significantly between the Zn-HAP and ZnSrMg-HAp groups (Figure 4D). The reason for this may be that the BV/TV was examined at an ROI of 1.1 mm and the animals were sacrificed at least 2 weeks after implant placement, as discussed previously. Compared with HAp coatings on titanium implants, the Zn-HAp and ZnSrMg-HAp groups showed significantly less bone mineral density (Figure 4E); these two groups exhibited more newly formed and unmineralized woven bone around the titanium implants, which in turn caused the lower bone mineral density.

## 4. Materials and Methods

### 4.1. Preparation of HAp, Zn-Doped HAp, and Zn-Sr-Mg-Doped HAp Coatings on Titanium Implants Using the VIPF-APS Technique

The preparation details of HAp, Zn-doped HAp, and Zn-Sr-Mg-doped HAp powders were described in a previous study [30]. Briefly, HAp coatings doped with different metal ions were prepared and denoted as HAp, Zn-HAp [Zn/(Ca+Zn) = 2.5%], and ZnSrMg-HAp [Zn/(Ca+Zn+Sr+Mg) = 2.5%, Sr/(Ca+Zn+Sr+Mg) = 5%, Mg/(Ca+Zn+Sr+Mg) = 5%]. The molar ratio of Ca(NO_3_)_2_/(NH_4_)_2_HPO_4_ was 10:6 for HAp, that of [Ca(NO_3_)_2_ + Zn(NO_3_)_2_]/(NH_4_)_2_HPO_4_ was 10:6 for Zn-HAp, and that of [Ca(NO_3_)_2_ + Sr(NO_3_)_2_ + Mg(NO_3_)_2_ + Zn(NO_3_)_2_]/(NH_4_)_2_HPO_4_ was 10:6 for ZnSrMg-HAp. The synthetic parameters were pH = 10 and a calcination temperature of 800 °C. The VIPF-APS technique was used to deposit HAp, Zn-HAp, and ZnSrMg-HAp on titanium implants (Stryker Leibinger GmbH & Co. KG, Freiberg, Germany) that were made of Ti6Al4V alloy, 5 mm in length and 1.2 mm in diameter (Figure 6). The technique involved water vapor being splashed on the titanium implant. The surfaces of the implant were roughened by sandblasting to Ra = 11.20 μm and then sprayed with a thin layer of HAp coating. After immersion in pure water, a single spray cycle was applied to the implant surface repeatedly until the desired thickness of HAp coating was achieved (30 µm). Finally, a porous surface structure was formed [33].

### 4.2. Reverse Transcriptase Quantitative Polymerase Chain Reaction

Human embryonic palatal mesenchymal (HEPM) cells (ATCC^®^CRL-1486TM) were incubated on titanium discs (diameter: 15 mm, thickness: 2 mm, Biomate Swiss GmbH, Zug, Switzerland) coated with HAp, Zn-HAp, or ZnSrMg-HAp for 3, 7, and 11 days. Subsequently, HEPM cells were used with TRIzol reagent (15596026, Invitrogen, Waltham, MA, USA) to extract mRNA. Purified cellular total RNA (100 ng) was used with the Power SYBR™ Green RNA-to-CT™ 1-Step Kit for reverse-transcription reaction. The corresponding cDNA was used in a quantitative polymerase chain reaction (qPCR) assay with Biorad CFX96 (Bio-Rad, Hercules, CA, USA) with primer sets of COL1A1, F: GATTCCCTGGACCTAAAGGTGC, and R: AGCCTCTCCATCTTTGCCAGCA; DCN, F: AGCTGAAGGAATTGCCAGAA, and R: CTCTGCTGATTTTGTTGCCA; TNFRSF11B, F: GAACCCCAGAGCGAAATAC, and R: CGCTGTTTTCACAGAGGTC; SPP1, F: CGAGGTGATAGTGTGGTTTATGG, and R: GCACCATTCAACTCCTCGCTTTC; and GAPDH, F: GTCTCCTCTGACTTCAACAGCG, and R: ACCACCCTGTTGCTGTAGCCAA [50,51].

### 4.3. Cell Culture, Protein Extraction, and Immunoblot Analysis

HEPM cells were cultured on titanium discs coated with HAp, Zn-HAp, or ZnSrMg-HAp for 7 and 11 days at 37 ℃ in an incubator with 5% CO_2_ in Dulbecco’s modified Eagle’s medium/F12 (1:1) (1×) (21041-025, Gibco™ - Thermo Fisher Scientific, Waltham, MA, USA) supplemented with 10% fetal bovine serum and 1% antibiotic pen-strep-amphotericin. HEPM cells were lysed using radioimmunoprecipitation assay buffer (W-7849-500, Goal Bio, Taipei, Taiwan) and cellular lysates were centrifuged at 12,000× *g* rpm for 5 min for supernatant collection. The extracted protein was quantified using a protein assay kit (500-0006, Bio-Rad, Hercules, CA, USA). Equal amounts of protein were separated using 10% sodium dodecyl sulfate-polyacrylamide gel electrophoresis and transferred to Amersham Hybond P 0.45 μm polyvinylidene fluoride (10600023, GE Healthcare, Chicago, IL, USA). After blocking with 5% skimmed milk, the membranes were incubated with various primary antibodies and then incubated with the corresponding secondary antibodies. The protein bands were detected using an Amersham ECL Select Western Blotting Detection Reagent (RPN2235, GE Healthcare, Chicago, IL) and quantified using ImageQuant 5.2 software (Healthcare Bio-Sciences, Pittsburgh, PA, USA). All experiments were repeated in triplicate (*n* = 3).

### 4.4. Antibacterial Activity Test

*P. gingivalis* (ATCC 33277) and *P. nigrescens* (ATCC 332563) were stored at −80 °C and separately cultured on Brucella blood agar plates (Taiwan Prepared Media, Taipei, Taiwan) at 37 °C for 7 days under standard anaerobic conditions (80% N_2_, 10% H_2_, and 10% CO_2_). A strain of a single colony of these bacteria was then separately cultured in 5 mL of brain heart infusion broth (Neogen, Lansing, MI, USA) with 5 g of yeast extract (Thermo Fisher, Waltham, MA, USA) and an L-cysteine solution (0.5 g/mL) (Sigma-Aldrich, Lyon, France) at 37 °C under anaerobic conditions for 48 h. Subsequently, these bacteria were collected via centrifugation at 3000× *g* rpm for 10 min. Each resultant bacterial pellet was washed 3 times with sterile phosphate-buffered saline and then adjusted to a concentration of 1.6 × 10^8^ colony-forming units (CFU)/mL before use [52]. The test samples were sterilized using saturated steam at 121 °C for 30 min. The HAp, Zn-HAp and ZnSrMg-HAp titanium discs were placed in *P. gingivalis* and *P. nigrescens* broth and incubated at 37 °C under anaerobic conditions for 120 h (*n* = 4). The bacterial suspension served as the control group. After 120 h, 1 mL was taken from each suspension, pipetted onto anaerobic blood agar, and cultured at 37 °C under standard anaerobic conditions for 120 h to count the CFU.

### 4.5. Animal Model

Eight-week-old male Sprague Dawley rats (The Jackson Laboratory, BioLASCO Taiwan Co., Ltd., Taipei, Taiwan, 232.11 ± 20.33 g) were used (*n* = 18, randomly divided into 6 groups in 6 cages) following the guidelines and protocols of the Institutional Animal Care and Use Committee of National Taiwan University (IACUC-20200127). All animals were given free access to water and standard rat food (fat 50 g/kg, protein 226 g/kg, and metabolizable energy 3030 kcal/kg). The environment was maintained at a temperature between 20 °C and 24 °C and relative humidity between 40% and 70%. Moreover, the rats were kept under a 12 h light/dark cycle. All animal experiments complied with the ARRIVE guidelines and were conducted following the National Research Council’s Guide for the Care and Use of Laboratory Animals.

### 4.6. Implantation

Thirty-six titanium implants were coated with HAp, Zn-HAp, and ZnSrMg-HAp (n = 12). The titanium implants were randomly allocated to 2 experiment durations (2 and 4 weeks, *n* = 6 each) and placed in the tibia of 18 rats that were anesthetized with 20–40 mg/kg of Zoletil^®^ 50 (Virbac, Carros, France) mixed with 5–10 mg/kg of Rompun^®^ (Bayer, Leverkusen, Germany). Each rat received a 5–6 mm longitudinal skin incision along the tibia. After soft tissue dissection, 2 holes were drilled into the tibia bone using a 1.0 mm electronic drill at low speeds (800–2000 rpm), and 3 types of titanium implants were randomly allocated to be inserted into the holes. The wound was sutured layer by layer, and the stitches were removed after 1 week of healing. In addition, analgesics and antibiotics were provided in the drinking water (0.2 mg/mL of ibuprofen and 0.268 mg/mL of ampicillin in 5% dextrose) to reduce post-surgical pain and infection. All rats were sacrificed after 2 and 4 weeks of implantation. A flowchart of the animal experiment is shown in Figure 7.

### 4.7. Micro-CT Evaluation

The tibia was harvested and scanned using the high-resolution SKYSCAN 1076 micro-CT apparatus (Skyscan NV, Kontich, Belgium). The bone coverage rate was defined as the percentage of direct bone-to-implant contact on the titanium implants. Two cylinders (0.85 and 1.1 mm in radius, 2.0 mm in length) starting from the first thread of the titanium implant were defined as the region of interest (ROI) [53,54]. For the evaluation of the bone volume fraction (bone volume/total volume, BV/TV) and bone mineral density (BMD), the ROI was defined as a cylinder (1.1 mm in radius and 2.0 mm in length) starting from the first thread of the titanium implant. The micro-CT technician and histological analyzer were blind to the sample groups.

### 4.8. Histological Examination

After fixation, serial dehydration, and embedding, resin blocks were trimmed to an appropriate size. The implant was sectioned into symmetrical halves using a low-speed saw (Isomet, Buehler Ltd., Lake Bluff, IL, USA) with a wafering diamond blade (6.6 cm × 0.15 mm) [55]. Cutting was performed at a blade speed of 100 to 500 rpm, using tap water as a lubricant, and with a force of 0.3 to 7 N acting on the specimen. The surface of the exposed specimen was polished using 600- or 800-grit silicon carbide paper followed by 4000-grit paper under water lubrication to remove cutting marks and obtain a highly polished surface. Subsequently, specimens were stained with Stevenel’s blue and then counterstained with alizarin red S for histological examination.

### 4.9. Statistical Analysis

An analysis of variance was used to examine the differences among all the groups, and Tukey’s post hoc test was used for qPCR and Western blotting to identify significant differences between 2 specific groups. A *p*-value less than 0.05 was considered statistically significant.

## 5. Conclusions

In conclusion, porous ZnSrMg-HAp was successfully coated on titanium implants using the VIPF-APS technique. The ZnSrMg-HAp group was the most effective at inducing mRNA and protein expression of TNFRSF11B and SPP1 after 7 days of incubation, and TNFRSF11B and DCN after 11 days of incubation. The ZnSrMg-HAp group also demonstrated superior antibacterial activity against *P. gingivalis*. The animal model suggested that BV/TV was significantly higher in the Zn-HAp and ZnSrMg-HAp groups at 2 and 4 weeks compared to the HAp group. In addition, continuous bone coverage and thickened bone deposition was more apparent on the implant surface in the ZnSrMg-HAp group than in the HAp and Zn groups. However, notably, bone adjacent to the implant surface does not equate to a structural or functional connection between bone and the implant. Both osteointegration and antibacterial properties are essential characteristics for preventing peri-implantitis. Therefore, ZnSrMg-HAp has the potential to be used as a bioactive coating material for implant surfaces and consequently improve the survival rate in clinical use.

## Figures and Tables

**Figure 1 ijms-24-04933-f001:**
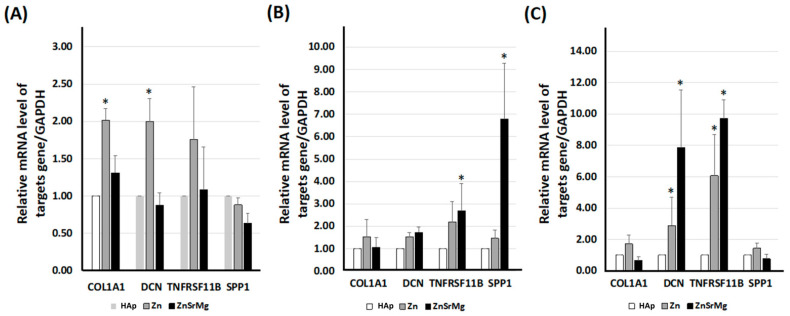
Expression of osteointegration-associated gene expression in HEPM. The HEPM cells were incubated on titanium discs coated with HAp, Zn-HAp, or ZnSrMg-HAp for 3 (**A**), 7 (**B**), and 11 days (**C**). Purified cellular mRNA was applied for a qPCR assay with primer sets of COL1A1, DCN, TNFRSF11B, and SPP1. * *p* < 0.05 vs. HAp group.

**Figure 2 ijms-24-04933-f002:**
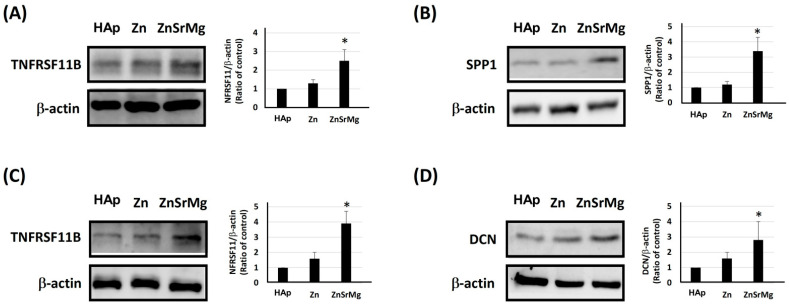
Expression of osteointegration-associated proteins in HEPM. The HEPM cells were incubated on titanium discs coated with HAp, Zn-HAp, or ZnSrMg-HAp for 7 (**A**,**B**) and 11 days (**C**,**D**). The cellular lysate was used for a Western blot assay with antibodies of TNFRSF11B (**A**,**C**), SPP1 (**B**), DCN (**D**), and β-actin (**A**–**D**). * *p* < 0.05 vs. HAp group.

**Figure 3 ijms-24-04933-f003:**
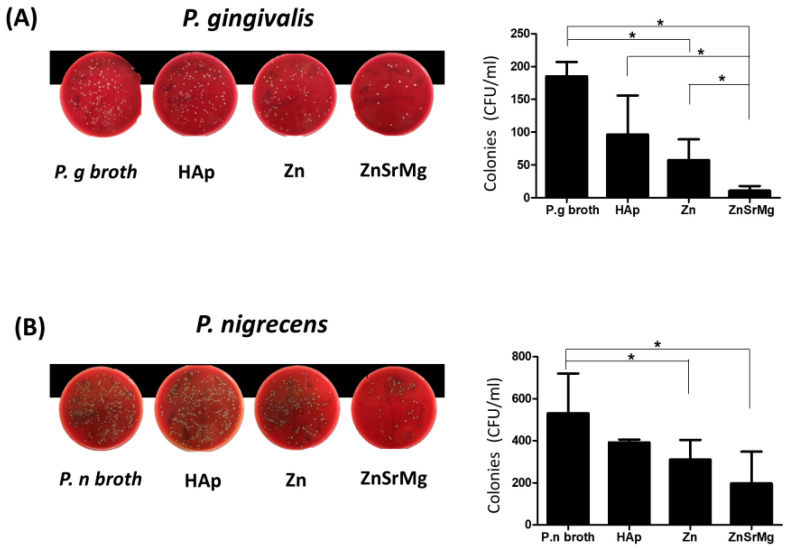
Antibacterial activity against *P. gingivalis* and *P. nigrescens* in the HAp, Zn-HAp, and ZnSrMg-HAp groups. (**A**) The growth of a *P. gingivalis* colony on Brucella blood agar plates (left). The quantification of *P. gingivalis* is shown as CFU/mL (right). (**B**) The growth of a *P. nigrescens* colony on Brucella blood agar plates (left). The quantification of *P. nigrescens* is shown as CFU/mL (right). CFU: colony forming unit, * *p* < 0.05.

**Figure 4 ijms-24-04933-f004:**
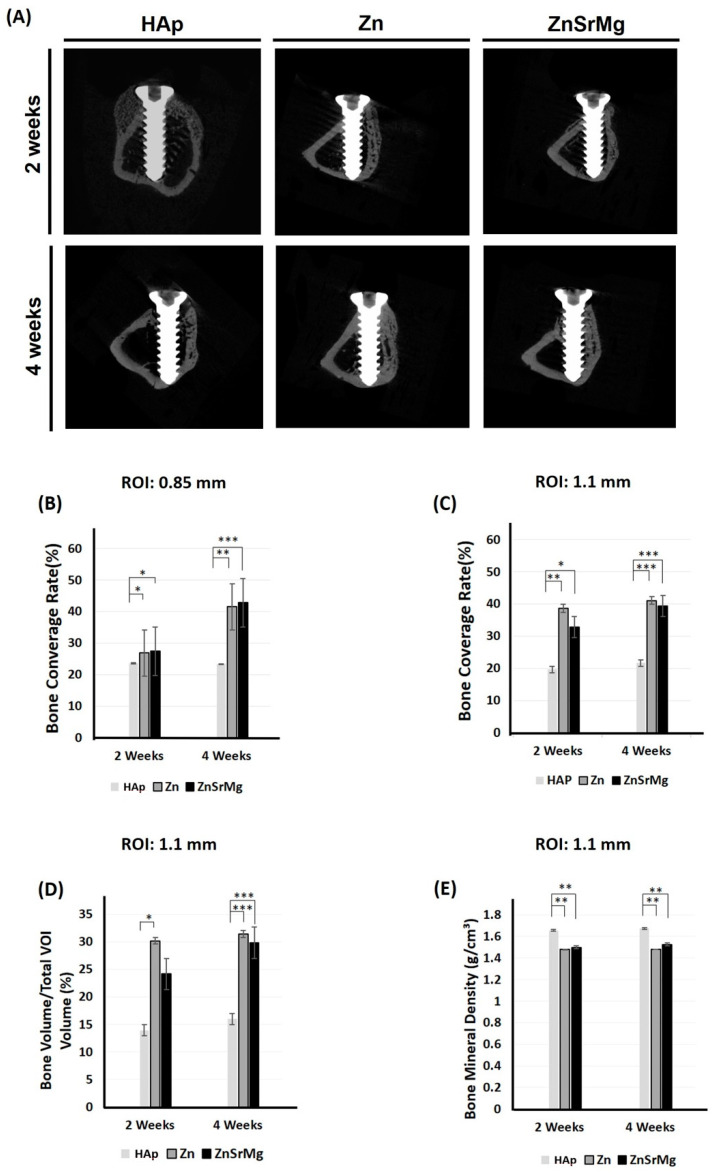
Measurement of new bone formation according to CT imaging. Micro-CT reconstruction images (**A**). New bone formation in the HAp, Zn-HAp, and ZnSrMg-HAp groups was recorded as bone coverage rate (%) (ROI: 0.85 mm (**B**); ROI: 1.1 mm (**C**)), BV/TV (%) (ROI: 1.1 mm (**D**)), and bone mineral density (g/cm^3^) (ROI: 1.1 mm (**E**)) at 2 and 4 weeks. ROI: region of interest, VOI: volume of interest. * *p* < 0.05, ** *p* < 0.01, *** *p* < 0.001.

**Figure 5 ijms-24-04933-f005:**
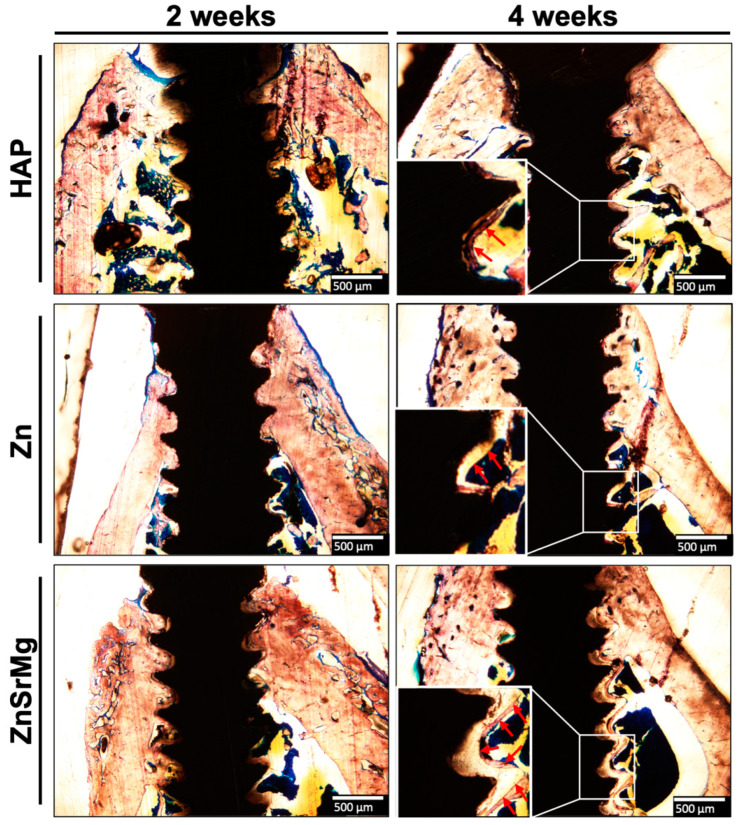
Histological examination of new bone formation using Stevenel’s blue and Alizarin red S. The histological results of the HAp, Zn-HAp, and ZnSrMg-HAp groups were collected at 2 and 4 weeks. (40× magnification). Continuous bone coverage on implant threads was more prominent in the ZnSrMg-HAp group, extending to the sixth thread, compared with the HAp and Zn groups. A pattern of concentrated bone growth could be clearly observed in the ZnSrMg-HAp group. Thickened bone deposition was more apparent on the implant surface in the ZnSrMg-HAp group. Red arrows indicate new bone formation (100× magnification).

**Figure 6 ijms-24-04933-f006:**
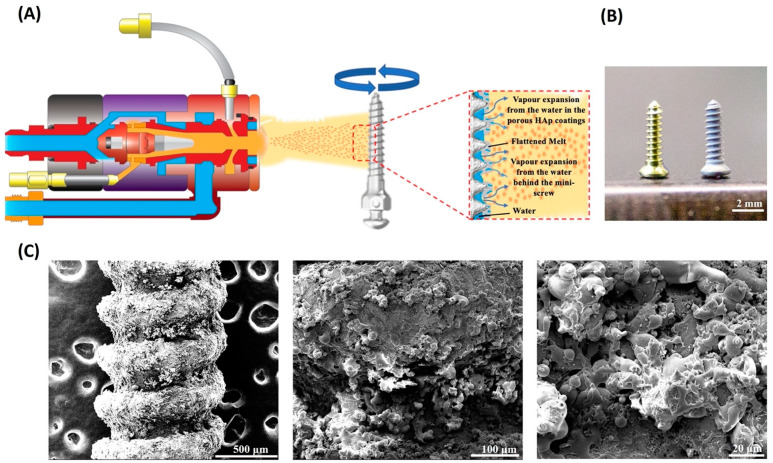
Preparation of HAp, Zn-doped HAp, and Zn-Sr-Mg-doped HAp coatings on titanium implants using the VIPF-APS technique. (**A**) The titanium implant was roughened and then sprayed with a thin layer of HAp coatings. After immersion in pure water, quick sprays were applied repeatedly, and the desired thickness of HAp coating was produced. (**B**) Left: titanium implants; right: titanium implants with the porous HAp coating. (**C**) Scanning electron microscopy images of HAp coatings on implant surfaces at different magnifications. (Magnification: 160×, 600×, 2500×; from left to right) VIPF-APS: vapor-induced pore-forming atmospheric plasma spraying.

**Figure 7 ijms-24-04933-f007:**
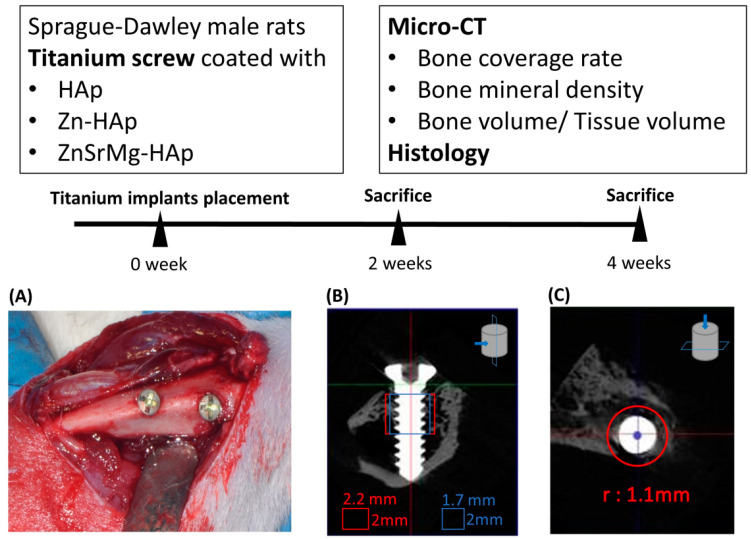
Flowchart of animal study and measurement of new bone formation. (**A**) Two titanium implants with different HAp coatings were placed in rat tibias. (**B**) For bone coverage rate, the ROI was set as a cylinder (0.85 and 1.1 mm in diameter and 2.0 mm in height) starting from the implant center. The radius of the titanium implant was 0.6 mm. (**C**) For BV/TV and bone mineral density, the ROI was defined as a cylinder (1.1 in diameter and 2.0 mm in height) starting from the first thread of the titanium implant. ROI: region of interest; BV/TV: bone volume/total volume.

## Data Availability

The data presented in this study are available on request from the corresponding author.

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
