# Peer review of "Vapor-Induced Pore-Forming Atmospheric-Plasma-Sprayed Zinc-, Strontium-, and Magnesium-Doped Hydroxyapatite Coatings on Titanium Implants Enhance New Bone Formation—An In Vivo and In Vitro Investigation"

_ijms, 2023, doi:10.3390/ijms24054933_

Round 1

Reviewer 1 Report (Previous Reviewer 2)

This is a very good article that should be recommended for publication, however few short comments should be taken in to account.

1.     1. The introduction is very detailed and reflects well the current state of the problems under consideration.

2.     Figure 1.  On parts A and B, it would be useful to show the rulers so that it is clear what the sizes are. Furthermore, the legend on the lower part of C is not visible well, so the quality of this drawing should be improved.

3.     In the conclusions, it is necessary to clearly formulate what new data about the studied problems were obtained in this work?

In its current form, the conclusion is rather short and does not reflect what has been done in this paper.

4.     Reference list.

Reference 1 – no page number.

Reference 8 – no article number.

Reference 36 – no article number.

Author Response

Reviewer 2 Report (Previous Reviewer 1)

The article presents zinc-, strontium- and magnesium- multidoped hydroxyapatite (HAp) porous coating to enhance new bone formation on titanium implants.  The paper is well-written, and the authors provided detailed results to assess the promotion of bone formation and the prominent osteogenesis described in the text. In  the abstract, the authors claimed:

"Furthermore, according to both micro-CT and histologic findings, the ZnSrMg-HAp group exhibited the most prominent osteogenesis and concentrated bone growth along implant threads. " However, reading carefully the results presented, I believe this claim is not the right one. Please review according to the authors´ observations from 387 to 409 lines. Apart from that, the rest of the work may be recommended for publication.

Author Response

This manuscript is a resubmission of an earlier submission. The following is a list of the peer review reports and author responses from that submission.

Round 1

Reviewer 1 Report

The article presents zinc-, strontium- and magnesium- multidoped hydroxyapatite (HAp) porous coating to enhance new bone formation on titanium implants.  The paper is well-written; however, the authors needed to provide all the results to assess the promotion of bone formation and the prominent osteogenesis described in the text.  Some critical issues are the following:

1) The micro-CT evaluation images and reconstructions were not included in the article.  This is mandatory in this study as the only use of the plots in Fig.6 is not enough.

2) In terms of mRNA expression, the authors did not include ALP, an early bone formation marker, and OCN, a late bone formation marker.  For 11 days, COL1A1 is a marker that supports many tissues in the body, including cartilage, bone, tendon, and skin.  The primary function of decorin DCN is regulation during the cell cycle.  What the authors suggest is happening does not seem to be an accurate picture of the mRNA expression study.

3) In Figure 4, the expression of osteointegration-associated proteins in HEPM does not seem significantly different between Zn-HAp and ZnSrMg-HAp in the images.  It is the analysis of the blots the one that provides these differences.  The original immunoblots of the replicates must be included in the supplementary material and better explain the study of the expression using the ImageQuant.  Also, it needs to be clarified the number of experiments in the statistical analysis.

4) The staining in Figure 7 is not correct.  The blue is not staining the ECM areas, and the AR does not seem specific, and I believe something happened when preparing the histologic micrographs.  Also, the authors' claims are not according to images.  For a proper evaluation, they should only focus on the interface between the implant and the coated implants to evaluate the test condition of new bone tissue.  Therefore, I'm afraid I have to disagree with the evaluation without more detailed images.

I recommend rejecting the paper for reasons I mention in this review related to the quality of the experimental work and the authors claims based on the results presented.

Other notes:

- In the introduction, there should be a paragraph about the texturing process of dental implants, such as https://doi.org/10.1016/j.cirp.2020.04.065, among many others.

- Include references in the sentence: "Various studies 48 have proposed increasing surface hydrophilicity and attracting osteoprogenitor cells using the advantages of metal ions."

- In methods, please include the references to the primer sets to evaluate their specificity.

- Ref [23] does not seem to be the one that the authors describe in the paragraphs.

Author Response

Responses to reviewers’ comments:

We appreciate the valuable comments from editor and reviewers and made the revision with point-by-point responses as follows. We are grateful to the reviewers who raised important issues and undoubtedly improved this manuscript. This manuscript has been undergone extensive English revisions by professional editors.

Reviewer 2 Report

Referee Report on “Vapor-induced pore-forming atmospheric plasma sprayed zinc- , strontium-, and magnesium-doped hydroxyapatite coatings on titanium implants enhance new bone formation – An in vivo and in vitro investigation

This is, of course, a work that could be recommended for publication, but only after some several improvements formulated below.

1.     In the Introduction, it was useful in more detail the latest trends in research and applications of HAP.  See, for your information:

https://www.mdpi.com/search?q=+%22hydroxyapatite%22&article_type=review-article

2.     For considered application of HAP, it is important to note that they are resistant to X-ray and UV irradiation without visible aging/ structural damage. See for example:

Bystrova, A.; Dekhtyar, Y.D.; Popov, A.; Coutinho, J.; Bystrov, V. Modified hydroxyapatite structure and properties: Modeling and synchrotron data analysis of modified hydroxyapatite structure. Ferroelectrics 2015, 475, 135–147.

Hübner, W.; Blume, A.; Pushnjakova, R.; et al. The influence of X-ray radiation on the mineral/organic matrix interaction of bone tissue: An FT-IR microscopic investigation. Int. J. Artif. Organs 2005, 28, 66–73.

This will give the article more visibility and interest from a wide range of readers.

3.     Regarding porosity, I would like to know how detailed the pore structure was studied, whether the pores are subject to temporal evolution and how their size affects the considered functional properties.

4.     List of the reference. The vast majority are quite old references, and in this regard, in the introduction, I would like to more clearly see the novelty and relevance of this work

Author Response

(The authors gave the same response as above.)

Reviewer 3 Report

The present work describes the synthesis method of Zn-, Sr- and Mg-multidoped HAp porous coatings on titanium discs and implants through the VIPF-APS technique to effectively improve the osteogenesis and antibacterial properties of the coatings. It is of interest to read this report, giving a great deal of characterization and performance evaluation. The paper is innovative, results are interesting and worth publishing, but the manuscript needs substantial improvements as follows:  

1. more discussion about the HAp composite should be conducted in the introduction part, and I would like to suggest the following literatures: “Biomineralization of bone-like hydroxyapatite to upgrade the mechanical and osteoblastic performances of poly(lactic acid) scaffolds”, International Journal of Biological Macromolecules https://doi.org/10.1016/j.ijbiomac.2022.11.240.

2. Section 2.1, the morphology and elemental characterization of the prepared Zn-HAp and ZnSrMg-HAp porous coatings should be provided to compare with HAp coating results.

3. Figure 1(A), the words on the image seem unclear. Please adjust the font size.

4. In section 2.7 of the Micro-CT evaluation, the ROI was defined as a cylinder (1.1 mm in radius and 2.0 mm in length) starting from the first thread of the titanium implant, why? and what was the basis for the selection?

5. Figure 7, the ROI of histological findings should be marked to improve the readability.

6. The format of references should be checked carefully to comply with this journal.

Author Response

(The authors gave the same response as above.)

Round 2

Reviewer 2 Report

After successful improvements, this manuscript can be recommended for publication